# Cholesterol-modified sphingomyelin chimeric lipid bilayer for improved therapeutic delivery

Zhiren Wang [1,5], Wenpan Li [1,5], Yanhao Jiang [1], Jonghan Park[1], Karina Marie Gonzalez[1], Xiangmeng Wu[1], Qing-Yu Zhang[1,2] & Jianqin Lu [1,2,3,4] ✉

Cholesterol (Chol) fortifies packing and reduces fluidity and permeability of the lipid bilayer in vesicles (liposomes)-mediated drug delivery. However, under the physiological environment, Chol is rapidly extracted from the lipid bilayer by biomembranes, which jeopardizes membrane stability and results in premature leakage for delivered payloads, yielding suboptimal clinic efficacy. Herein, we report a Chol-modified sphingomyelin (SM) lipid bilayer via covalently conjugating Chol to SM (SM-Chol), which retains membrane condensing ability of Chol. Systemic structure activity relationship screening demonstrates that SM-Chol with a disulfide bond and longer linker outperforms other counterparts and conventional phospholipids/Chol mixture systems on blocking Chol transfer and payload leakage, increases maximum tolerated dose of vincristine while reducing systemic toxicities, improves pharmacokinetics and tumor delivery efficiency, and enhances antitumor efficacy in SU-DHL-4 diffuse large B-cell lymphoma xenograft model in female mice. Furthermore, SM-Chol improves therapeutic delivery of structurally diversified therapeutic agents (irinotecan, doxorubicin, dexamethasone) or siRNA targeting multi-drug resistant gene (p-glycoprotein) in late-stage metastatic orthotopic KPC-Luc pancreas cancer, 4T1-Luc2 triple negative breast cancer, lung inflammation, and CT26 colorectal cancer animal models in female mice compared to respective FDA-approved nanotherapeutics or lipid compositions. Thus, SM-Chol represents a promising platform for universal and improved drug delivery.

Liposome (Lipo), composed of lipid bilayer(s) comprising phospholipid(s) and cholesterol (Chol), has been successful for packaging and delivering therapeutic agents due to its intrinsic biocompatibility and biodegradability[1–4]. While most FDA-approved liposomal nanotherapeutics can improve pharmacokinetics and ameliorate side effects, improvements in therapeutic efficacy and overall survival are limited even for the best nanoformulations and completely missing for majority[5,6], underscoring the urgent need of an improved platform for enhanced therapeutic delivery. Chol plays a critical role in fortifying membrane packing and reducing bilayer fluidity and permeability by promoting the liquid condensed state in lipid membranes, enhancing bilayer rigidity and strength[7–10].

[1]Skaggs Pharmaceutical Sciences Center, Department of Pharmacology & Toxicology, R. Ken Coit College of Pharmacy, The University of Arizona, Tucson, AZ 85721, USA. [2]Southwest Environmental Health Sciences Center, The University of Arizona, Tucson 85721, USA. [3]Clinical and Translational Oncology Program (CTOP), The University of Arizona Cancer Center, Tucson, AZ 85721, USA. [4]BIO5 Institute, The University of Arizona, Tucson, AZ 85721, USA. [5]These authors contributed equally: Zhiren Wang, Wenpan Li. ✉e-mail: lu6@arizona.edu

Lipid bilayers with a high percentage of Chol are generally more stable than those with less Chol[11]. Nevertheless, Chol can be readily transferred between biomembranes and lipoproteins under physiological conditions[12–14], which sabotages liposomal stability and results in premature contents leakage, subsequent fast blood clearance, and unwanted systemic adverse effects, resulting in disappointing therapeutic efficacy in clinic[5,6]. To tackle this key bottleneck in liposomal drug delivery, an improved lipid bilayer that forms Lipo but cannot shuttle between biomembranes to cement drug packaging and therapeutic delivery is ideal. Previously, Dr. Francis C. Szoka, Jr.'s team developed a series of sterol-modified lysophospholipids (SMLs: PChcPC, PChemsPC, OChemsPC, DChemsPC) via covalently attaching Chol to single aliphatic chain lysophospholipids using ester and carbonate ester bonds[15–17]. Their results showed that the SML Lipo effectively reduced Chol exchange and decreased content leakage in 30% FBS compared to phospholipids/Chol Lipo. However, doxorubicin-loaded SML Lipo had less tumor uptake and did not improve antitumor efficacy compared to Doxil[15]. In addition, Sergelius et al. constructed a N-cholesteryl sphingomyelin by replacing the amide-linked acyl chain with cholesterol carbamate, which enhanced bilayer order, conferring resistance against detergent solubilization[18]. Chol contains a hydroxyl group, which allows conjugation with various functional groups or therapeutic agents to facilitate drug and gene delivery using Lipo. 3β-[N-(N′,N′-dimethylaminoethyl)-carbamoyl]cholesterol (DC-Chol) was developed as a cationic Lipo reagent for gene therapy and vaccine delivery system[19–22]. Based on this, another two cationic lipids were constructed via modifying Chol with positively charged basic amino acid residues (lysine and histidine) to enhance gene delivery efficiency[23]. In addition, Chol-derivatization strategy was applied to drugs (eg., Paclitaxel-7-carbonyl-cholesterol, Tax-Chol) that cannot be readily loaded into Lipo[24]. Chol can also be modified with targeting peptides to improve the targeting efficiency of Lipo[25–27]. In these Chol derivatives, Chol functioned as an anchor to insert into the lipid bilayer. Although these works were meaningful, they did not improve the physicochemical properties of the lipid bilayer and the additional chemical groups attached to Chol can potentially jeopardize the bilayer stability[28].

Of note, the phospholipids present in the cell membrane contain a double aliphatic chain[29]. To further improve the therapeutic delivery efficiency of Lipo and better mimic the cell membrane composition, we proposed to engineer an improved lipid bilayer through covalently conjugating sphingomyelin (SM) that has a double aliphatic chain and is one of the core phospholipids in the cell membrane to Chol. SM was chosen as the model phospholipid because 1) it is a naturally occurring phospholipid in mammalian cell membrane and has a hydroxyl group (Fig. 1a) that enables its conjugation with Chol[30–32], and 2) it is a backbone component in liposomal vincristine (VCR) nanomedicine, Marqibo[33]. We posited that the SM-Chol would impart several advances over prior SMLs and conventional phospholipids/Chol systems through enhancing lipid bilayer cohesion property and subsequently improving drug encapsulation and delivery. First, the amide linkage in SM is less susceptible to physiological degradation compared to the ester bonds in lysophospholipids in SMLs and other double aliphatic chain phospholipids, leading to enhanced bilayer stability[29]. Second, the amide bond in SM-Chol provides a hydrogen bond donor, which enables the formation of the intermolecular and intramolecular hydrogen bonding, boosting the bilayer stability. In stark contrast, this cannot be achieved in SMLs as which do not possess free hydrogen bond donors. Third, the double aliphatic chain SM-Chol increases bilayer compressibility and decreases permeability to water in comparison to the single aliphatic chain SML[29,34]. Fourth, to control and selectively trigger the bilayer dissociation for timely drug release, apart from the ester and carbonate ester bonds used in SMLs, we leveraged varied stimuli-responsive bonds (e.g., cathepsin B (glycine bond), glutathione (disulfide bonds), and reactive oxygen species (ROS,

thioketal bond) present in inflammatory diseases and cancers) with distinct linker chemistry to bridge SM with Chol (Fig. 1a).

In this work, SM-Chol well retains the membrane condensing ability of Chol (Fig. 1i–l, and Supplementary Fig. 25). Systemic structure activity relationship screening reveal that Lipo composed of SM-Chol with a disulfide bond and longer linker (SM-CSS-Chol) perform better than other conjugates, SMLs and traditional phospholipids/Chol Lipo on blocking Chol transfer and preventing payload leakage (Fig. 1f–h); and increase maximum tolerated dose (MTD) of vincristine while diminishing systemic toxicities (Fig. 2), improve pharmacokinetics and tumor delivery efficiency, and boost tumor reduction and prolong mice survival in SU-DHL-4 diffuse large B-cell lymphoma xenograft model (Fig. 3). Further, SM-CSS-Chol bilayer fortifies drug packaging and therapeutic delivery of other therapeutic agents (irinotecan (IRI), doxorubicin (DOX), dexamethasone (DEX)) with diverse chemical structures in late-stage metastatic orthotopic KPC-Luc pancreas cancer, 4T1-Luc2 triple negative breast cancer, and lung inflammation mouse models in comparison with respective FDA-approved nanotherapeutics (Onyvide, Doxil, Figs. 4–6) and SML Lipo. In addition to small molecule drugs, SM-CSS-Chol can also enhance the gene delivery efficiency of the siRNA targeting p-glycoprotein (P-gp), the common drug efflux pump, compared to SM/Chol and SML when both integrating the FDA-approved ionizable lipid, Dlin-MC3-DMA (DMA, Fig. 6h–k)[35]. These findings substantiate that SM-Chol boasts favorable and improved biophysicochemical properties over conventional phospholipids/Chol and SML systems, revealing its potential for improved liposome-based therapeutic delivery.

## Results
### Development of the Chol-derived SM delivery platform
To prevent the Chol exchange and investigate the impact of diverse linker chemistry bridging SM and Chol, five different SM-Chol conjugates were designed and synthesized (Fig. 1a and Supplementary Figs. 1–5) – one with a carbonate ester bond (SM-C-Ester-Chol), one with ester bond (SM-Ester-Chol), one with glycine bond (SM-Glycine-Chol), one with disulfide bond and a longer linker (SM-CSS-Chol), and one with thioketal bond with a longer linker (SM-SCS-Chol), which can be cleaved by high levels of hydrolase, cathepsin B, glutathione (GSH), or reactive oxygen species, respectively, in cancers and inflammatory diseases[36–40]. The synthesized SM-Chol conjugates were verified by [1]H NMR, [13]C NMR, and ESI-MS (Supplementary Methods and Supplementary Figs. 1–5, 12–16). Four SMLs (PChcPC, PChemsPC, OChemsPC and DChemsPC purchased from Avanti Polar Lipids) were used as controls. The Lipo formed by SM-Chol chimeric membrane were similar to SM/Chol Lipo (Lipo-SM/Chol) concerning the dynamic light scattering (DLS) size, zeta potential and stability (Fig. 1b–d and Supplementary Figs. 20, 24). To evaluate the impact of disulfide exchange on Chol release from the SM-CSS-Chol and the release of encapsulated drugs at the tumor/disease sites, we assessed the payload release kinetics using calcein (a fluorescent dye) and SM-CSS-Chol remained in Lipo-SM-CSS-Chol over time in PBS or glutathione (GSH)[41]. The calcein release was very low and well controlled in PBS (pH = 7.4), which was readily accelerated in the presence of GSH (Supplementary Fig. 21). SM-CSS-Chol was quite stable in PBS (pH = 7.4) while being degraded rapidly when GSH was present (Fig. 1e, Supplementary Figs. 21, 22). Further studies demonstrated that the sulfide-linked sphingomyelin (intermediate 7, Supplementary Fig. 23) was produced and was confirmed by high-resolution LC-MS (HRMS) in the presence of GSH, which corroborated that Lipo-SM-CSS-Chol was GSH-responsive due to the disulfide linkage. To assess the osmotic stress-induced bilayer structural deformation, we measured the leakage profiles of SM-Chol under a high osmotic gradient in comparison to various conventional phospholipids (SM, HSPC, SPC, DSPC, or DOPC)/Chol and SMLs (PChcPC, PChemsPC, OChemsPC, and DChemsPC) by using calcein as a model payload. CSS and SCS bridged SM-Chol

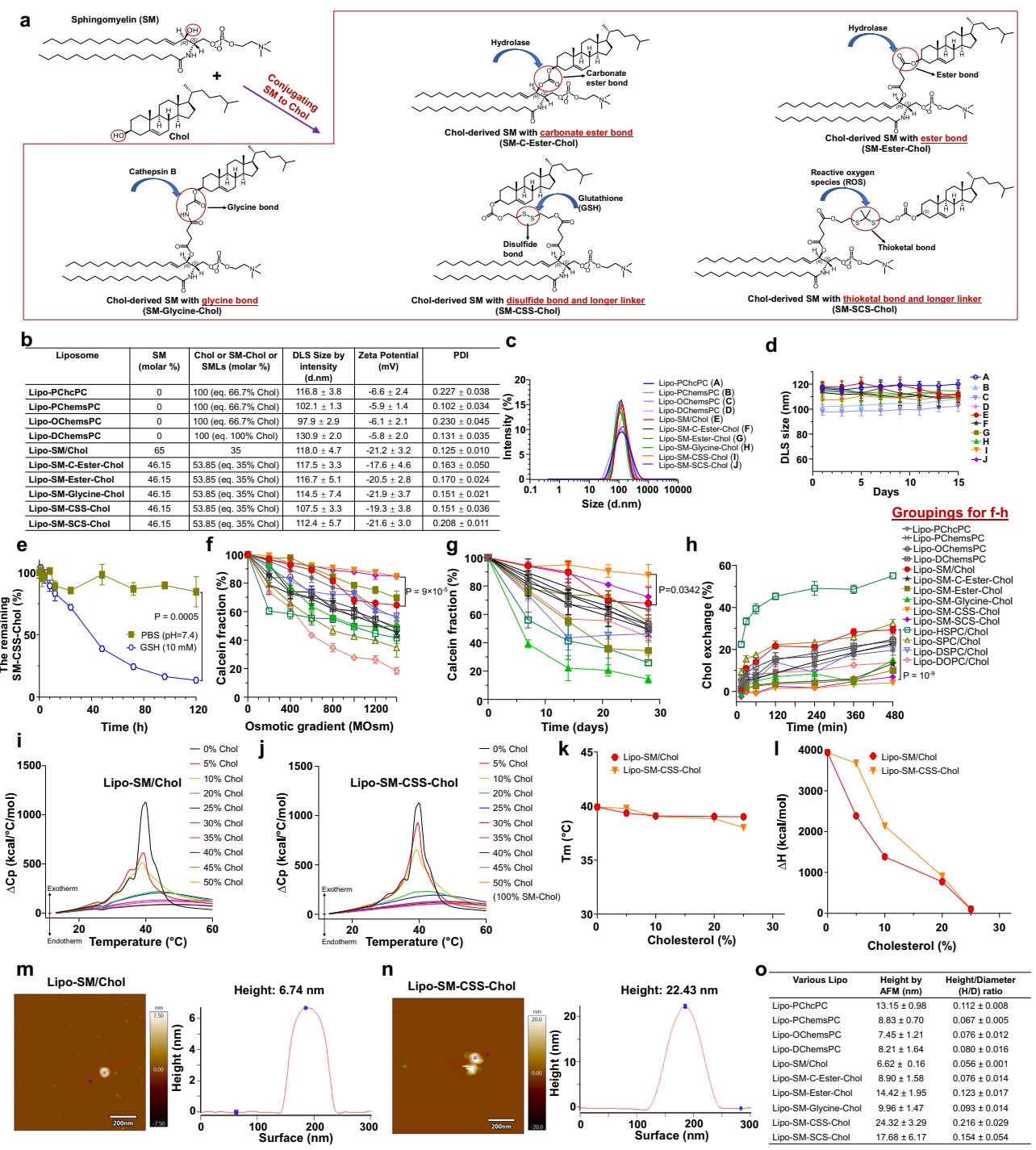

**Fig. 1 | Development of SM-derived Chol Lipo (Lipo-SM-Chol). a** Synthesis of SM-Chol conjugate with a carbonate ester bond (SM-C-Ester-Chol), an ester bond (SM-Ester-Chol), a glycine bond (SM-Glycine-Chol), a disulfide bond (SM-CSS-Chol) with a longer linker or thioketal bond (SM-SCS-Chol) with a longer linker. **b** A table depicting the physicochemical characterizations of Lipo composed of sterol-modified phospholipids (SMLs) from Avanti Polar Lipids with equivalent (eq.) 66.7 mol % Chol, SM/Chol or five SM-Chol with equivalent (eq.) 35 mol % Chol. d.nm, diameter values in nanometers. **c** DLS size distribution by intensity for Lipo-SM/Chol and Lipo-SM-Chol, DLS: dynamic light scattering. **d** The monitoring of the DLS size by intensity over time in 5% dextrose at 4 °C. **e** The percentage of remaining SM-CSS-Chol measured by LC-MS/MS after incubated Lipo-SM-CSS-Chol with or without the presence of GSH at 37 °C. **f, g** Leakage profiles for calcein-loaded Lipo (eq. 40 mol % Chol) in high osmotic gradient (**f**) or 30% fetal bovine serum (**g**) at 37 °C. **h** Relative Chol exchange rates at 37 °C with eq. 40 mol % Chol in the Lipo., **i-l** Thermotropic phase transition behavior determined by differential scanning calorimetry (DSC). DSC thermograms of Lipo-SM/Chol (**i**) and Lipo-SM-CSS-Chol (**j**) at various eq. mol % Chol. The effects of Chol or SM-CSS-Chol on transition temperature (**k**) and enthalpy (**l**). **m–o** Atomic force microscopy (AFM) to assess the height of Lipo-SM/Chol (**m**) and Lipo-SM-CSS-Chol (**n**) and the lipid bilayer rigidity was shown as by the ratio of height/diameter (H/D) value (**o**)[43] (n = 3 independent experiments, similar results were observed). Data in **b, d, e–h, o** are expressed as mean ± s.d. (n = 3 independent experiments). Statistical significance was determined by one-way ANOVA followed by Tukey's multiple comparisons test, two-tailed, unpaired Student's t-test for **e**. Source data are provided as a Source Data file.

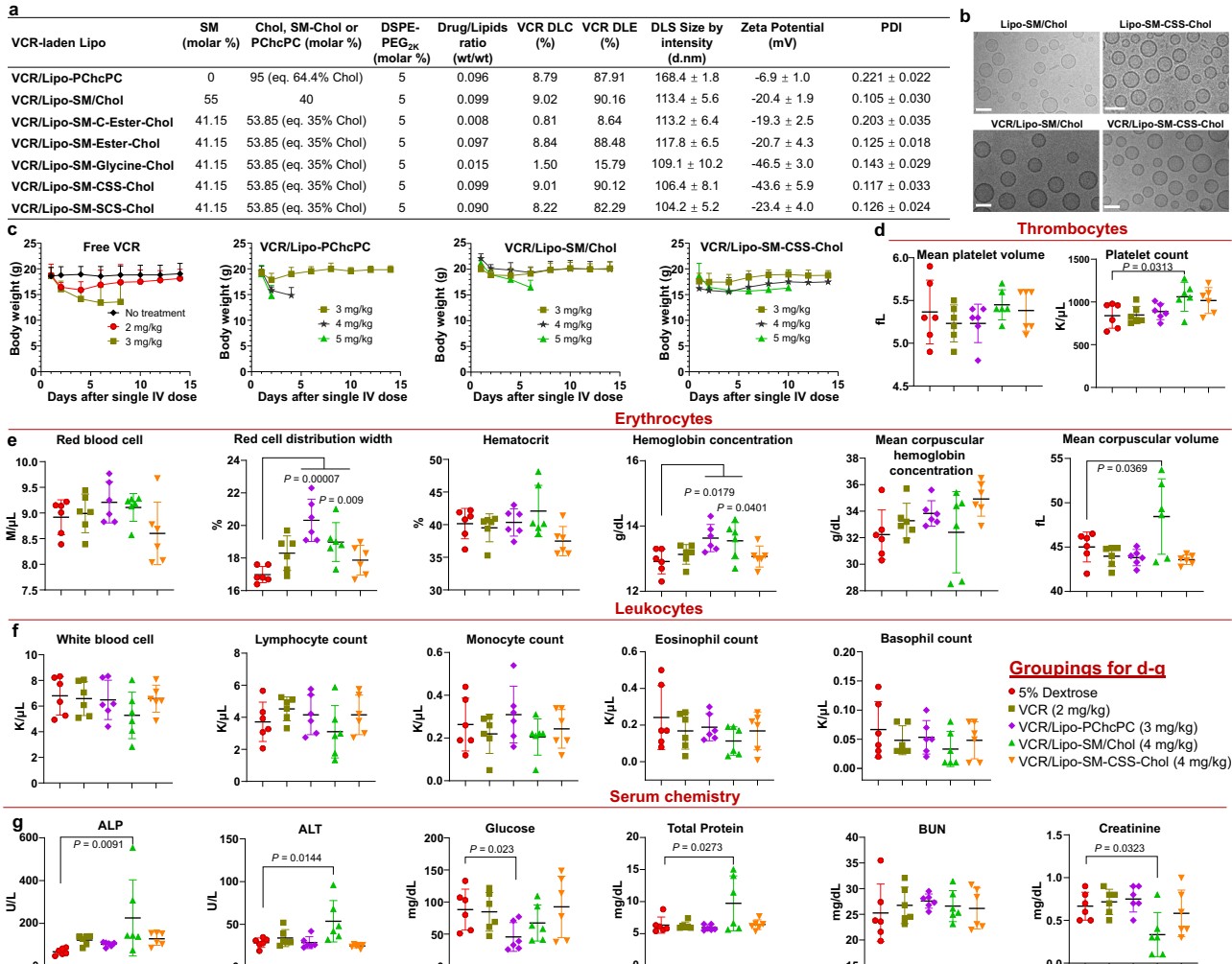

**Fig. 2 | VCR-laden Lipo-SM-Chol increased the maximum tolerated dose (MTD) of VCR without systemic toxicities in healthy mice. a** A table delineating the physicochemical characterizations of various VCR/Lipo with eq. 35 mol % Chol (40% Chol for Lipo-SM/Chol to match the ratio used in Marqibo) and 5% DSPE-PEG[2K]. DLC: drug loading capacity; DLE: drug loading efficiency. **b** Cryo-electron microscopy (cryo-EM) of Lipo-SM/Chol or Lipo-SM-CSS-Chol with or without VCR encapsulation. Scale bar: 100 nm, (n = 3 independent experiments, similar results were observed). **c** The mice weight monitoring in MTD study of free VCR, VCR/SML Lipo (Lipo-PChcPC), VCR/Lipo-SM/Chol, and VCR/Lipo-SM-CSS-Chol at various

doses as indicated in healthy C57BL/J mice following a single i.v. administration via tail vein; Mice body weight and survival were monitored for 2 weeks. The MTD was defined by the dose that did not cause mouse death or more than 15% weight loss during the whole period[36,84]. **d**–**g** On day 14 post i.v. injection, blood was withdrawn for comprehensive thrombocytes (**d**), erythrocytes (**e**), leukocytes (**f**), and serum chemistry (**g**) analysis. Data in **a** (right portion, n = 3 independent experiments), **c**–**g** (n = 6 mice) are expressed as mean ± s.d. Statistical significance was determined by one-way ANOVA followed by Tukey's multiple comparisons test. Source data are provided as a Source Data file.

performed better than other SM-Chol counterparts, SMLs controls, and other phospholipid/Chol systems particularly in Lipo-SM-CSS-Chol (Fig. 1f). To evaluate the impact of physiological environment (biomembranes) to extract free Chol to induce content leakage, the payload retention in Lipo-SM-Chol was determined in 30% fetal bovine serum (FBS). SM-CSS-Chol outperformed other SM-Chol, SMLs, and various phospholipid/Chol systems in content retention with minimal leakage (Fig. 1g). Additionally, our Chol exchange study unveiled that SM-Chol bridged by CSS or SCS bonds had much lower Chol transfer than SMLs and phospholipid/Chol mixtures, corroborating Chol-derived SM can block the Chol exchange between biomembranes more efficiently (Fig. 1h). Adding free Chol into the bilayer composed of phospholipids has proven to modify the thermotropic phase behavior of the bilayer and the phase transition can be eliminated at certain mol % Chol, yielding a solid lipid phase[42]. To define the influence of SM-Chol on the phase transition of SM, we used differential scanning calorimetry (DSC). DSC thermograms showed the thermotropic phase transition of SM was eliminated when mixed with 30 mol

% free Chol. SM-Chol exhibited a similar pattern as free Chol, rendering the phase transition of SM disappeared at eq. 30 mol % Chol (40 mol % for SM-SCS-Chol) and decreased transition temperature (Tm) and enthalpy ($\Delta H$) in a SM-Chol-dependent manner (Fig. 1i–l and Supplementary Fig. 25), substantiating covalently attaching Chol to SM retained the Chol membrane condensing ability. Further, atomic force microscopy (AFM) revealed that SM-Chol bilayer had a much higher degree of stiffness than SM/Chol membrane as evidenced by higher height/diameter (H/D) ratio (Fig. 1o and Supplementary Fig. 27)[43]. Given the importance of cholesterol and sphingomyelin in biological membranes and in cells, we investigated whether intracellular delivery of Lipo-SM-Chol affects Chol and SM trafficking via evaluating the sterol regulatory element-binding protein 1 (SREBP1, a key transcriptional factor that controls lipogenesis and lipid uptake)[44,45], and Niemann Pick C (NPC) proteins (responsible for intracellular Chol transport)[46], as well as cell membrane lipid rafts formation and SM levels[47,48]. Our western blot proved that Lipo-SM-Chol had no effect on SREBP1 and NPC1 and NPC2 proteins compared

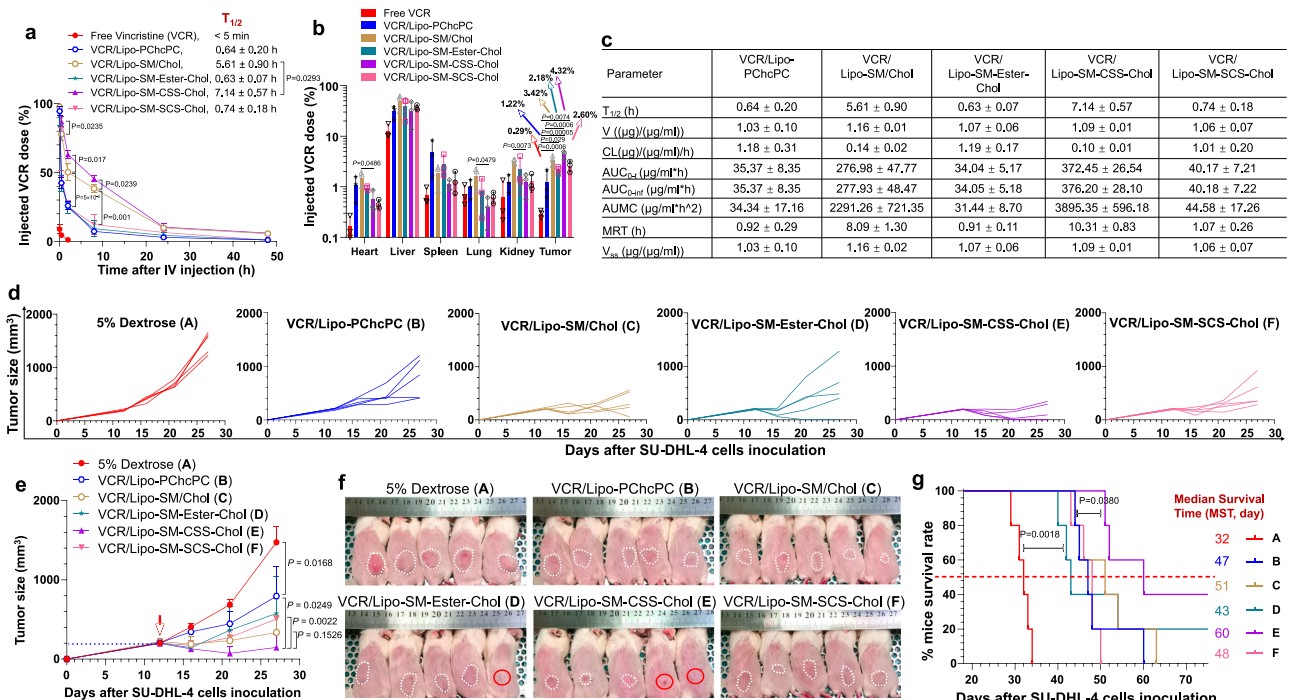

**Fig. 3 | Improved circulation time, tumor delivery and therapeutic efficacy of VCR/Lipo-SM-CSS-Chol. a–c** Blood kinetics (**a**), biodistribution (**b**, at 48 h post i.v. injection) and pharmacokinetic parameters (**c**) of free VCR and VCR/Lipo in orthotopic MC38 colorectal cancer (CRC) mouse model (n = 3 mice; tumor: ~400 mg) following a single i.v. administration at 2 mg VCR/kg. **d** Individual tumor growth curve. **e** Average tumor growth curve. **f** Mice bearing s.c. lymphoma image taken on day 27. **g** Kaplan–Meier survival curves. Data in **a–c** (n = 3 mice), **e** (n = 5 mice) are expressed as mean ± s.d. Statistical significance was determined by one-way ANOVA followed by Tukey's multiple comparisons test; survival curves were compared using the log-rank Mantel–Cox test. Source data are provided as a Source Data file.

immunodeficient CB17/Icr-*Prkdc^scid*/IcrIcoCrl mice i.v. injected once at 2 mg VCR/mg. **d** Therapeutic effects of VCR/Lipo in subcutaneous (s.c.) SU-DHL-4 diffuse large B-cell lymphoma xenograft model (n = 5 mice, tumors: ~200 mm³) in severe combined

with the vehicle control (Supplementary Fig. 29). Moreover, confocal laser scanning microcopy showed that the levels of lipid rafts[49,50] and SM[51,52] were not significantly altered on cells treated with Lipo-SM-Chol compared to vehicle control (Supplementary Figs. 30, 31). Taken together, these data demonstrated that Lipo-SM-Chol did not affect the intracellular trafficking of Chol and SM. To elucidate if the unique structure of SM-Chol causes toxicity, we evaluated its cytotoxicity in 4T1 triple-negative breast cancer cells. All Lipo-SM-Chol had no significant cell-killing activity at up to 1 mM and were as well-tolerated as diverse other phospholipid/Chol mixtures (Supplementary Fig. 28).

## MTD, pharmacokinetics and anti-lymphoma effects of VCR-laden Lipo

Given SM/Chol Lipo is used to deliver VCR in Marqibo, we first examined the efficiency of SM-Chol in delivering VCR. Using the same lipids ratio and remote loading strategy with citrate buffer as the pH gradient as in Marqibo[53], our in-house made VCR/Lipo-SM/Chol well resembled that of commercial Marqibo regarding the drug loading capacity (DLC)/efficiency (DLE) (Supplementary Fig. 33), in which the VCR content was determined by High-Performance Liquid Chromatography (HPLC, Supplementary Fig. 32), nanoparticle size, zeta potential, polydispersity (PDI) and morphology (Fig. 2a, b)[54,55]; Our VCR/Lipo-SM-Chol with Ester, CSS, or SCS linkage showed similar characterizations as SM/Chol. Of note, C-Ester and Glycine bonded SM-Chol displayed much lower DLC and DLE compared to SM/Chol and other SM-Chol conjugates (Fig. 2a, b), which is attributable to their relatively poor leakage profiles (Fig. 1f, g), indicating the linker chemistry played a significant role in defining the physicochemical properties of SM-Chol membrane. Then, we evaluated the MTD of VCR/Lipo-SM-CSS-Chol and VCR/Lipo-SM/Chol, both of which had similar DLC/DLE with identical drug/lipids ratio (wt/wt = 0.099, Fig. 2a and Supplementary

Fig. 33), in comparison to free VCR and VCR/Lipo-PChcPC in healthy C57BL/J mice. Different does of free VCR and VCR Lipo at a single intravenous (i.v.) injection were investigated. Consistent with literature[53,56], free VCR had the MTD at 2 mg/kg. VCR/Lipo-PChcPC increased the MTD to 3 mg/kg, while Lipo-SM/Chol further elevated it to 4 mg/kg (Supplementary Fig. 34). Lipo-SM-CSS-Chol also exerted the VCR MTD to 4 mg/kg. Of note, VCR/Lipo-SM/Chol caused abnormal alkaline phosphatase (ALP), alanine transaminase (ALT), total protein, creatinine, red cell distribution width, heamoglobin concentration, and mean corpuscular volume levels and VCR/Lipo-PChcPC exhibited abnormal red cell distribution width, heamoglobin concentration and glucose, however, these were not seen in VCR/Lipo-SM-CSS-Chol (Fig. 2c–g). These findings substantiate the excellent in vivo safety profile of VCR/Lipo-SM-CSS-Chol and the ability to maximize the therapeutic potential.

To delve deeper into the in vivo stability and therapeutic delivery efficiency of SM-Chol, we systemically explored the blood kinetics and biodistribution in orthotopic MC38 colorectal cancer (CRC) tumor model. Our data have shown that within 5 min, ~90% free VCR was cleared from the blood stream. In sharp contrast, VCR/Lipo-SM-Chol markedly extended the circulation half-life of VCR and delivered 6.5- to 13.9-fold more VCR into tumor. These effects were more significant in Lipo-SM-CSS-Chol which was also superior to Lipo-SM/Chol and Lipo-PChcPC (Fig. 3a–c). Notably, Lipo-SM-CSS-Chol has significantly less distribution to heart, lung, and kidney than VCR/Lipo-SM/Chol, allowing it to further minimize the systemic adverse effects (Fig. 3b). In addition, we systemically evaluated the in vivo stability and payload release in tumors of various Lipo in orthotopic KPC-Luc pancreas tumor mouse model via encapsulating MU-P into the core of Lipo and incorporating DiD, a far-red fluorescent dye into the lipid bilayer[17], which enabled the tracking of the Lipo and its content (Supplementary

Fig. 37). The ratio of the total exposure of MU-P to the total exposure of lipid ($AUC_{MU-P}/AUC_{DiD}$) can determine how stably Lipo retain the contents during circulation. Our MU-P/DiD/Lipo-SM-CSS-Chol had greatly higher $AUC_{MU-P}/AUC_{DiD}$ ratio (0.86) than that of other MU-P/DiD/Lipo-SM-Chol (0.43-0.46), MU-P/DiD/Lipo-PChcPC (0.69), and MU-P/DiD/Lipo-SM/Chol (0.29), revealing its superior in vivo stability. We confirmed that Lipo-SM-CSS-Chol showed slower rate of content delivery in the liver, spleen, and kidney tissues compared to Lipo-SM/Chol, Lipo-PChcPC, and other Lipo-SM-Chol. We also showed that Lipo-SM-CSS-Chol exhibited a much higher content release rate in tumors (higher conversion of MU-P to the MU and MU-G)[17] than Lipo-PChcPC, Lipo-SM/Chol and other Lipo-SM-Chol counterparts. This could be attributed to the higher GSH levels in tumor cells[41,57-60], which triggered the efficient dissociation of the lipid bilayer to allow rapid cargo release. The improved pharmacokinetics and tumor delivery, and increased payload release in tumors are crucial for enhanced therapeutic activity.

Since VCR is approved for treating diffuse large B-cell lymphoma (DLBCL)[61], thus we investigated the therapeutic efficacy of VCR/Lipo in human SU-DHL-4 DLBCL xenograft model in CB17/Icr-*Prkdc*^*scid*/IcrI-coCrl mice[62] (Fig. 3d–g) with an aim to see whether the Lipo-SM-Chol works better than Lipo-SM/Chol on controlling lymphoma development. Mice-bearing lymphoma were intravenously injected once by various VCR/Lipo at 2 mg VCR/kg when s.c. tumor reached ~200 mm³. S.c. DLBCL tumors grew rapidly in vehicle control-treated mice, reflecting its aggressive attribute. VCR/Lipo-SM/Chol was able to significantly reduce the tumor burden, demonstrating the advantage of using Lipo-SM/Chol as the drug carrier (Fig. 3d–f). Interestingly, VCR/Lipo-SM-Ester-Chol and VCR/Lipo-SM-SCS-Chol exhibited similar level of lymphoma suppression as VCR/Lipo-SM/Chol (Fig. 3d–f). Piggybacked by the higher improvements on pharmacokinetics and tumor delivery efficiency, VCR/Lipo-SM-CSS-Chol further enhanced the lymphoma tumor growth inhibition and eradicated 2 out of 5 lymphoma tumors in mice (Fig. 3d–f). Besides, similar results were discerned for the mice survival, where VCR/Lipo-SM-CSS-Chol imparted the longest mice survival rate (Fig. 3g). Since VCR works by binding to the tubulin, we measured β-tubulin via immunofluorescence imaging in SU-DHL-4 diffuse large B-cell lymphoma treated by various VCR-loaded Lipo in an independent study (Supplementary Fig. 38). We observed that the microtubule network was characterized by regularly assembled, normal filiform microtubules wrapped around the cell nucleus and detected the well-organized and bipolar mitotic divisions in tumor cells treated by vehicle control (Supplementary Fig. 38). In contrast, the microtubule spindles shrunk around the center of cells and the multipolarization of the spindle and multinucleation phenomena were suppressed after VCR/Lipo therapies, among which VCR/Lipo-SM-CSS-Chol showed the strongest effects. Moreover, VCR/Lipo-SM-CSS-Chol exerted the highest level of enhancing apoptosis (cleaved caspase-3, CC-3) and DNA breaks (terminal deoxynucleotidyl transferase dUTP nick end labeling, TUNEL) as well as reducing cell proliferation (Ki67) (Supplementary Fig. 38).

## Lipo-SM-CSS-Chol fortifies therapeutic delivery of IRI and DOX in metastatic PDAC and TNBC

To elucidate whether Lipo-SM-Chol can efficiently encapsulate and deliver drugs with different chemical structures and polarity, we packaged IRI or DOX into the core of Lipo-SM-Chol in comparison to Lipo-SM/Chol and Lipo-PChcPC following the remote loading approaches used in corresponding FDA-approved liposomal nanotherapeutics Onivyde (TEA₈SOS solution as the pH gradient)[63] or Doxil (ammonia sulfate as the pH gradient)[64] and compared their anti-tumor activity with Onivyde or Doxil in late-stage metastatic orthotopic PDAC or TNBC model, respectively.

We observed that IRI-laden Lipo-SM-Chol with C-Ester or Ester bonds had similar DLC and DLE (Supplementary Fig. 40) as those of Lipo-SM/Chol and Lipo-PChcPC (the IRI content was measured by HPLC, Supplementary Fig. 39), which were further increased in Lipo-SM-Chol with CSS or SCS linkages, particularly in Lipo-SM-CSS-Chol (Fig. 4a). Since IRI is approved for treating PDAC in clinic, we established a stringent Kras and Trp53 mutated metastatic orthotropic KPC-Luc (LSL-Kras^G12D/+;LSL-Trp53^R172H/+;Pdx-1-Cre, with luciferase expression) murine PDAC model to mimic the human PDAC because KPC-Luc reproduces many of the key features of the tumor microenvironment as seen in human PDAC[65,66]. Within 11 days post inoculating KPC-Luc cells into the pancreas of mice, the primary tumors grew to ~400 mg with significant metastasis to other organs (Fig. 4b). Vehicle control (VC, 5% dextrose) had no effect on controlling the tumor development and two mice died on day 20 and 24 (Fig. 4c), respectively, demonstrating the aggressiveness and invasiveness of this tumor type. IRI/Lipo-SM/Chol showed some tumor reduction and reduced the metastasis compared to VC, revealing the benefit of using a nanocarrier for the therapeutic delivery of IRI; Onivyde, the FDA-approved liposomal IRI, was able to further heighten the therapeutic effects (Fig. 4c–g). Interestingly, IRI/Lipo-SM-CSS-Chol outperformed against Onivyde, IRI/Lipo-SM/Chol, IRI/Lipo-PChcPC and other IRI/Lipo-SM-Chol counterparts and produced the highest level of KPC-Luc tumor inhibition with drastically diminished tumor metastasis to other organs (Fig. 4c–g).

Furthermore, to decipher the importance of the disulfide linkage, we have synthesized a SML in which the Chol is anchored to the 1-palmitoyl-2-hydroxy-sn-glycero-3-phosphocholine via a disulfide bond (SML-SS, Supplementary Fig. 10) as an additional control. Furthermore, to accomplish a more stringent comparison, a SML with the same disulfide linker (CSS) used in SM-CSS-Chol was also synthesized (SML-CSS, Supplementary Fig. 11) and used as another SML control. In addition, all the commercially available SMLs (PChcPC, PChemsPC, OChemsPC, and DChemsPC, purchased from Avanti Polar Lipids) were included as additional controls. Our data showed that IRI/Lipo-SMLs markedly reduced tumor growth with mitigated metastasis compared to vehicle control, especially in IRI/SML-SS, corroborating the advantage of using a disulfide bond linker (Supplementary Fig. 42). Interestingly, SML-CSS performed better than SML-SS on therapeutic delivery of IRI, which is attributed to the longer circulation time and higher tumor distribution (Supplementary Fig. 42b–d). Notably, our IRI/Lipo-SM-CSS-Chol is superior to all sterol-modified lipids controls on antitumor efficacy by garnering significantly more tumor reduction and minimizing tumor metastasis. The enhanced efficacy of IRI/Lipo-SM-CSS-Chol could be due to the improved pharmacokinetics and tumor delivery, and upregulated intratumoral γ-H2AX, CC-3, and attenuated Ki67 levels (Supplementary Fig. 42j).

Like Lipo-SM/Chol and Lipo-PChcPC, Lipo-SM-Chol enabled efficient packaging of DOX as evidenced by similar physicochemical characterizations concerning the DLC and DLE (Supplementary Fig. 45, the DOX content was determined by HPLC as published)[36], size, PDI, etc (Fig. 5a). Given that DOX is the standard of care for metastatic breast cancer treatment, we investigated the anti-tumor efficacy of diverse DOX/Lipo in metastatic orthotopic 4T1-Luc2 TNBC, as which resembles closely human breast cancer and is an ideal animal model for stage IV human breast cancer[67]. On day 15 after injecting 4T1-Luc2 cells into the 4th mammary fat pad, mice developed primary tumors ~200 mm³ and were intravenously administered by a single dose of DOX/Lipo. Doxil, the FDA-approved liposomal DOX, was used as the control. Tumor in VC-treated mice grew rapidly and metastasized to lung severely on day 35 (Fig. 5c–f and Supplementary Fig. 46). While DOX/Lipo-SM-Ester-Chol significantly reduced tumor burden and mitigated the lung metastasis compared to VC, DOX/Lipo-SM/Chol and DOX/Lipo-SM-SCS-Chol further boosted the therapeutic efficacy. However, these anti-TNBC effects were further augmented by Doxil treatment, reflecting the effectiveness of this clinically used

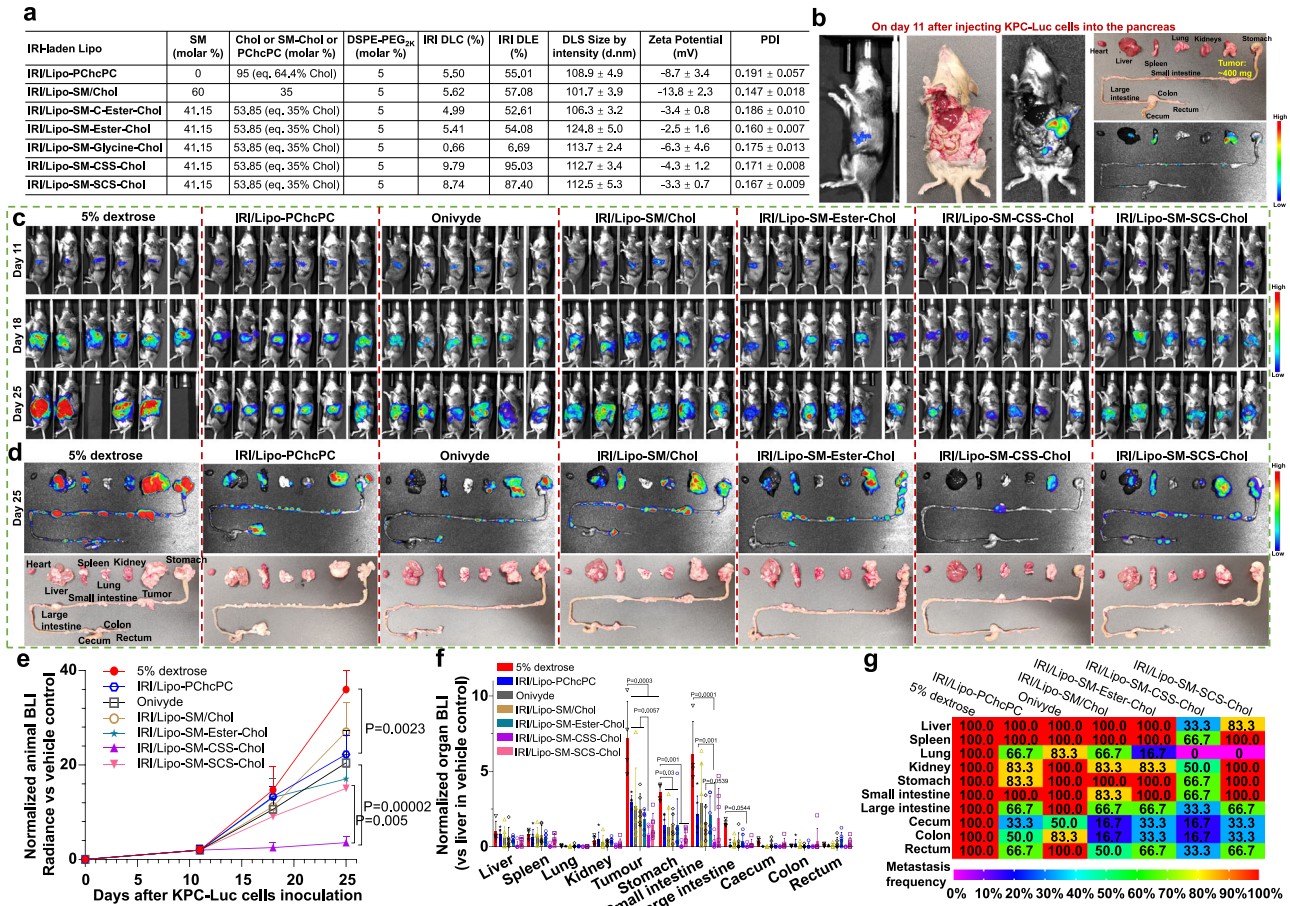

**Fig. 4 | Lipo-SM-CSS-Chol enhanced the therapeutic effects of IRI in late-stage metastatic orthotopic KPC-Luc pancreatic ductal adenocarcinoma (PDAC).**
**a**, A table showing the physicochemical characterizations of various IRI/Lipo with eq. 35 mol % Chol (eq. 64.4 mol % Chol for IRI/Lipo-PChcPC)[15,17] and 5 mol % DSPE-PEG$_{2K}$ (n = 3 independent experiments). **b**–**g** Therapeutic efficacy in metastatic orthotopic PDAC tumor mouse model. A total of $2 \times 10^6$ cells were injected into the pancreas of B6129SF1/J mice (n = 6 mice). On day 11, the primary tumors reached ~400 mg with noticeable metastasis (**b**) and mice were intravenously injected with various IRI/Lipo or Onivyde at 40 mg IRI/kg on days 11, 14, and 17. **c** Mice Lago

bioluminescence imaging (BLI) on days 11, 18, and 25. Two mice in group A died on day 20 and 24, respectively. **d** Representative ex vivo BLI (upper panel) and photographs (lower panel) for various organs on day 25. **e** Normalized BLI for whole mice tumor burden. Normalized BLI in various organs (**f**) and a heatmap summarizing tumor metastatic rate (**g**) on day 25. Data in **a** (right portion), **e**, **f** (n = 6 mice) are expressed as mean ± s.d. Statistical significance was determined by one-way ANOVA followed by Tukey's multiple comparisons test. Source data are provided as a Source Data file.

nanoformulation. Strikingly, DOX/Lipo-SM-CSS-Chol was superior to Doxil, DOX/Lipo-PChcPC, and other groups on controlling tumor growth by shrinking tumor mass to around half of its starting point and prevented lung metastasis completely (Fig. 5c–f).

Through inhibiting topoisomerase I (IRI) or topoisomerase II (DOX), IRI and DOX can induce DNA damage. Based on the established literature, we examined the γH2AX, a sensitive molecular marker of DNA Damage via immunohistochemistry (IHC)[68–72]. Compared with vehicle control, IRI/Lipo or DOX/Lipo upregulated the γ-H2AX signal, especially in IRI/Lipo-SM-CSS-Chol or DOX/Lipo-SM-CSS-Chol treated groups in KPC-Luc and 4T1-Luc2 tumors, respectively (Supplementary Figs. 41, 47). Additionally, we also evaluated the CC-3, TUNEL, and Ki67, in which Lipo-SM-CSS-Chol markedly outperformed other Lipo-SM-Chol, Lipo-SM/Chol, Lipo-PChcPC, and Onivyde or Doxil on augmenting apoptosis and DNA breaks, as well as inhibiting cell proliferation when delivering IRI or DOX.

**Lipo-SM-CSS-Chol enhances the delivery of DEX to inflamed lung**
In addition to packaging chemotherapeutics (VCR, IRI, DOX) in the lumen, we attempted to investigate if Lipo-SM-Chol can enhance the therapeutic delivery for hydrophobic drugs via direct encapsulation into the lipid bilayer through thin-film hydration method[73]. To test this

hypothesis, DEX, a potent anti-inflammatory drug, was utilized as the model hydrophobic payload. We demonstrated that all Lipo-SM-Chol can improve the DLE for DEX (Supplementary Fig. 49a, the DEX content was measured by HPLC, Supplementary Fig. 48), particularly with Ester, CSS, and SCS-bonded SM-Chol, which resulted in 4 to 5.2-fold increase for DLE (Fig. 6a). The DEX/Lipo remained stable within a 2-week monitoring period (Supplementary Fig. 49b, c). In order to test the anti-inflammatory effects of DEX/Lipo, we established a murine lung inflammation model through intratracheally administering LPS into the lungs of BALB/c mice (Fig. 6b)[10,74]. 6 h after injecting LPS to mice, the mice were then intravenously treated with one dose of free DEX or DEX/Lipo. 12 h later, the lung tissues were isolated and processed to determine the IL-6, TNF-α, and IL-1β levels using enzyme-linked immunosorbent assay (ELISA). As depicted in Fig. 6c–e, no treatment group (with LPS injection) markedly elevated IL-6, TNF-α, and IL-1β levels as compared to the sham group (no LPS injection), manifesting the successful establishment of the inflamed lung model. Free DEX had no discernable effect on attenuating the IL-6, TNF-α and IL-1β cytokines, while DEX delivered by Lipo-SM/Chol or Lipo-PChcPC exhibited significant IL-6, TNF-α, and IL-1β reduction in lungs, proving the advantage of using Lipo to boost the therapeutic delivery to inflamed tissue[75]. Strikingly, DEX/Lipo-SM-CSS-Chol further alleviated

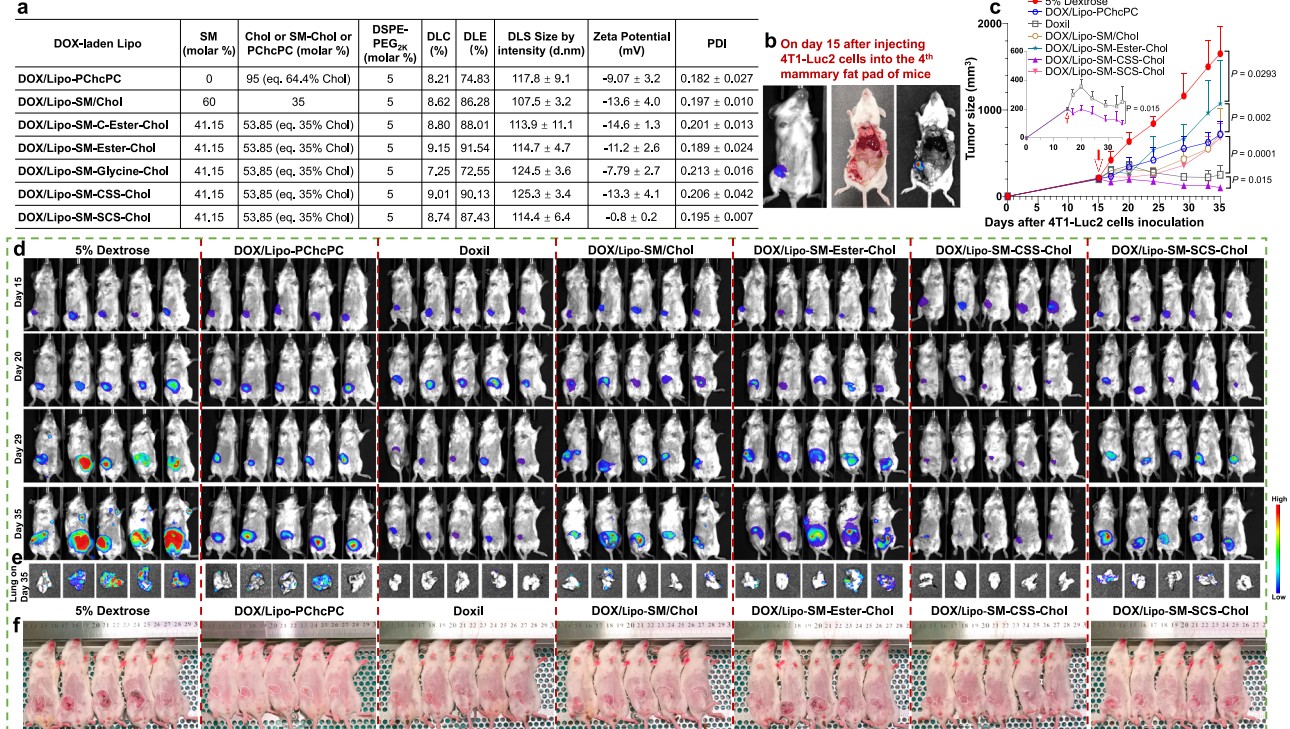

**Fig. 5 | Lipo-SM-CSS-Chol fortified the therapeutic delivery of DOX in orthotopic 4T1-Luc2 triple negative breast cancer (TNBC).** **a** A table delineating the physicochemical characterizations of DOX/Lipo with eq. 35 mol % Chol (eq. 64.4 mol % Chol for DOX/Lipo-PChcPC) and 5 mol % DSPE-PEG$_{2K}$ (n = 3 independent experiments). **b–f** anti-TNBC effects in metastatic orthotopic 4T1-Luc2 tumor mouse model. A total of $2 \times 10^5$ cells were injected into the 4th mammary fat pad of BABL/c mice (n = 5 mice)[82]. On day 15, the mice with primary tumors ~200 mm³ (**b**) received an i.v. administration of various DOX/Lipo or Doxil at 15 mg DOX/kg. **c** Average tumor growth curves measured by a digital caliper. **d** mice BLI on day 15, 20, 29, and 35 by Lago optical imaging. Ex vivo lung metastasis BLI from all 5 mice in each group (**e**) and tumor-bearing mice images (**f**) were taken on day 35. Data in **a** (right portion), **c** (n = 5 mice) are expressed as mean ± s.d. Statistical significance was determined by one-way ANOVA followed by Tukey's multiple comparisons test; survival curves were compared using the log-rank Mantel–Cox test. Source data are provided as a Source Data file.

lung inflammation by diminishing IL-6, TNF-α, and IL-1β to the next level (Fig. 6c–e). Apart from increased pro-inflammatory cytokines, peribronchial thickening and leukocyte recruitment are hallmarks in inflamed lungs[10]. To dive deeper into the efficacy of DEX/Lipo on these histopathological alterations in lung inflammation, we stained the lung sections with hematoxylin and eosin (Fig. 6f). We showed that mice treated with Lipo encapsulated DEX, particularly DEX/Lipo-SM-CSS-Chol, drastically inhibited the leukocyte recruitment and peribronchial thickening whereas mice treated with non-capsulated DEX did not prevent the peribronchial thickening and leukocyte recruitment into the lungs (Fig. 6f). These findings uphold that Lipo-SM-CSS-Chol worked better than Lipo-SM/Chol and Lipo-PChcPC system in delivering DEX to inflamed lungs.

## SM-CSS-Chol boosts the gene delivery efficiency of P-gp siRNA

In addition to small molecule therapeutic agents, we were interested in understanding whether SM-Chol can also improve the gene delivery efficiency in vivo. We chose to test the siRNA targeting the P-gp gene (*Abcb1a*) because P-gp is a formidable drug efflux pump in a variety of diseases including cancers and inflammatory diseases and has been plaguing a wide array of small molecule drugs (e.g., IRI, VCR, DOX, etc), yielding multi-drug resistance and poor therapeutic efficacy in the long run[76–79]. To deliver siRNA, we leveraged the ionizable lipid, Dlin-MC3-DMA (DMA) used in FDA-approved siRNA lipid nanoparticle (LNP), Onpattro[1] along with SM-CSS-Chol, PChcPC or SM/Chol. The lipid compositions/ratio (DMA/DSPC/Chol) used in Onpattro[35] were utilized as the control system for P-gp siRNA delivery. Using gel retardation assay, we found that free siRNA rapidly degraded after 8 h in serum; nonetheless, siRNA packaged in LNP-DMA/SM/Chol, LNP-

DMA/PChcPC or LNP-DMA/SM-CSS-Chol entailed siRNA serum stability for up to 24 h, which was in line with the siRNA/LNP-DMA/DSPC/Chol control (Fig. 6g). Afterwards, we investigated the in vivo P-gp gene knockdown efficiency of siRNA/LNP in CT26 CRC murine tumor model that has high expression of P-gp[80]. Via qRT-PCR, we found that siRNA/LNP-DMA/DSPC/Chol control decreased the P-gp mRNA level significantly in tumors, which was comparable to that of siRNA/LNP-DMA/SM/Chol and siRNA/LNP-DMA/PChcPC (Fig. 6h). Noteworthily, the gene silencing efficiency was further prominently enhanced in siRNA/LNP-DMA/SM-CSS-Chol. To elucidate if the improved P-gp siRNA delivery has real impact on intratumoural drug uptake and antitumor efficacy, we treated the mice bearing CT26 tumor with IRI/Lipo-SM-CSS-Chol plus siRNA/LNP-DMA/SM-CSS-Chol in comparison to IRI/Lipo-SM-CSS-Chol alone. Three times of i.v. injections of IRI/Lipo-SM-CSS-Chol led to marked tumor reduction, which was further increased when it was combined with siRNA/LNP-DMA/SM-CSS-Chol (Fig. 6i, j). While IRI/Lipo-PChcPC plus siRNA/LNP-DMA/PChcPC also enhanced the anticancer efficacy compared to IRI/Lipo-PChcPC, which was markedly outperformed by the combination of IRI/Lipo-SM-CSS-Chol plus siRNA/LNP-DMA/SM-CSS-Chol. The improved anticancer efficacy over IRI/Lipo alone was attributed to the enhanced drug delivery efficiency to tumors as combing siRNA/LNP significantly heightened IRI concentrations in tumors (Fig. 6k).

To address the common challenge, Chol exchange between biomembranes under physiological conditions, facing the liposome-based drug delivery, we coined an innovative chimeric lipid bilayer consisting of a Chol-derived SM. We discovered that covalently attaching Chol to SM securely confined Chol in the bilayer and retained the membrane-condensing capability of Chol. Systemic SAR screening

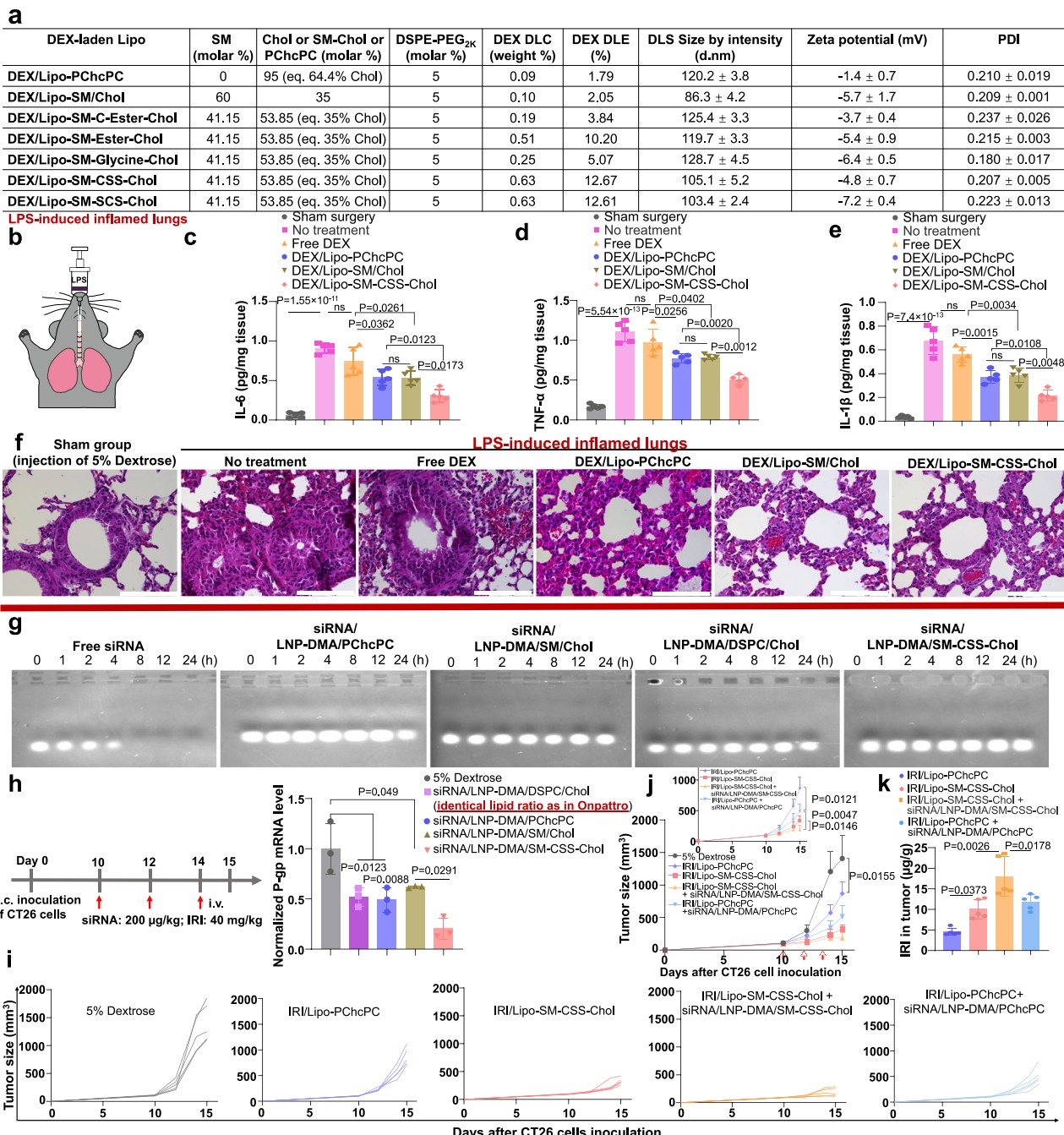

**Fig. 6 | Therapeutic efficacy of DEX-laden/Lipo-SM-Chol in lipopolysaccharide (LPS)-induced lung inflammation model (a-f), and improved gene delivery efficiency of siRNA-encased LNP--SM-Chol in silencing the multi-drug resistant P-gp gene in colorectal cancer (CRC) tumor model (g-k). a** a table showing the physicochemical characterizations of diverse DEX-laden Lipo with eq. 30 mol % Chol (eq. 64.4 mol % Chol for DEX/Lipo-PChcPC) and 5 mol % DSPE-PEG$_{2K}$ (n = 3 independent experiments). **b** the lung inflammation model was established by inoculating LPS (30 μL, 400 μg/mL) into the trachea of mice[10,74]; 6 h later, mice (n = 5 mice) were intravenously administered with one dose of free DEX or various DEX/Lipo at 1 mg DEX/kg. 12 h after treatment, 5 independent lung tissues were collected for pro-inflammatory cytokines: interleukin-6 (IL-6) (**c**), tumor necrosis factor-α (TNF-α) (**d**) and interleukin-1β (IL-1β) (**e**) examination[10,85]. Representative hematoxylin and eosin staining images from 5 independent lung tissues in each group (**f**). Scar bar = 100 μm, (n = 5 independent experiments, similar results were observed). **g** Serum stability analysis of free siRNA or siRNA/LNP (mixed with PBS,

v/v = 1:1, incubated at 37 °C) by gel retardation assay with 1% agarose gel electrophoresis[86], (n = 3 independent experiments, similar results were observed). **h** CT26 CRC tumor model (n = 3 mice) was established by s.c. injection of 1 × 10⁵ cells to mice. On day 10, when tumors reached ~100 mm³, mice were intravenously injected by different siRNA/LNP formulations (at eq. 39 mol % Chol and 1.5 mol % PEG$_{2K}$-C-DMG) at 200 μg P-gp siRNA/kg on day 10, 12 and 14. On day 15, tumors were collected for P-gp mRNA analysis by qRT-PCR. **i–k** in another parallel efficacy study, mice-bearing CT26 tumors (n = 5 mice; tumors: ~100 mm³) received three i.v. injections (on day 10, 12, and 14) of IRI/Lipo-SM-CSS-Chol or its combination with siRNA/LNP-DMA/SM-CSS-Chol. **i**, individual tumor growth curves. **j**, average tumor growth curves. **k**, on day 15, tumors were isolated for HPLC analysis to measure the IRI intratumoral uptake levels. Data in **a** (right portion), **c–e**, **h**, **j**, **k** are expressed as mean ± s.d. Statistical significance was determined by one-way ANOVA followed by Tukey's multiple comparison test for **c–e**, **h**, **k** or two-tailed, unpaired Student's t-test for **j**. Source data are provided as a Source Data file.

demonstrated that disulfide-bonded SM-Chol with a longer linker (SM-CSS-Chol) was superior to other SM-Chol conjugates, previous SMLs and many commonly used traditional phospholipids/Chol mixtures on blocking the Chol exchange and preventing payload leakage, indicating the linker chemistry played a significant role in defining the physicochemical properties of SM-Chol membrane. Furthermore, Lipo-SM-CSS-Chol improved the payload's pharmacokinetics and enhanced the therapeutic delivery of various drugs that have distinct molecular structures (VCR, IRI, DOX, DEX) and nucleic acid therapeutics (P-gp siRNA) in diverse disease animal models (DLBCL, Kras/Trp53-mutated metastatic PDAC, metastatic TNBC, lung inflammation, and CRC) in comparison to SM/Chol, PChcPC, SML-SS and SML-CSS and/or respective FDA-approved nanomedicines or lipid compositions. Our findings support that the SM-Chol can serve as a universal platform for improved drug and gene delivery to enhance the therapy and prevention of various human diseases.

## Methods

### Ethical statement
This research complies with all relevant ethical regulations. The animals were maintained under pathogen-free conditions and all animal experiments were approved by The University of Arizona Institutional Animal Care and Use Committee (IACUC).

### Cells culture
CT26 (Cat. ATCC CRL-2638) and 4T1 (Cat. CRL-2539) cell lines were obtained from UACC; 4T1-Luc2 (Cat. CRL-2539-LUC2) cells were purchased from ATCC; SUDHL4 (Cat. CRL-2957) cell was provided by Professor Catharine Smith at The University of Arizona; these cell lines were cultured in complete RPMI-1640 medium. MC38 (Cat. ENH204-FP) was purchased from Kerafast; KPC-Luc (Cat. 153474) was provided by Professor Gregory Beatty at the University of Pennsylvania; these cell lines were cultured in a complete DMEM medium. All the cell lines were cultured in the corresponding medium containing 10% FBS, 100 U/mL penicillin, 100 μg/mL streptomycin, and 2 mM L-glutamine at 37 °C in a $CO_2$ incubator.

### Animal assay
CB17/Icr-$Prkdc^{scid}$/IcrIcoCrl (Charles Rivers, 6 weeks old, female), and BALB/c, C57BL/6 and B6129SF1/J mice (Jackson laboratory, ~6 weeks old, female) were used. Standard Individually Ventilated Caging (IVC) system was used to maintain the mice under pathogen-free conditions. The animal house was kept at a temperature of 68–72°F and indoor humidity of 30-70% to abide by the NIH Guide and in accordance with the guidelines of 12 h light/12 h dark by 7 am on–7pm off. Digital caliper was utilized to measure the length and width of the tumor, and the formula = $0.5 \times \text{length} \times \text{width}^2$ was used to calculate the tumor size. The maximal permitted tumor size was 2000 $mm^3$ according to the animal ethics guidelines of IACUC and animal welfare regulations, and the mice were sacrificed once the tumor volume grew to ≥ 2000 $mm^3$ or the status of the mice became moribund. Nevertheless, the tumor size of some mice has grown greater than 2000 $mm^3$ by the final day of measurement, and the mice were sacrificed subsequently.

### Preparation of Lipo-SM-Chol
SM, Chol, SM-Chol conjugate or sterol-modified phospholipids (SMLs: PChcPC, PChemsPC, OChemsPC, DChemsPC), were dissolved in ethanol with a 100 mL round bottom glass flask. The solvent was evaporated under a rotatory evaporator (RV 10 digital, IKA®) to generate a thin film, which was further dried under ultra-high vacuum (MaximaDry, Fisherbrand) for 0.5 h. The film was hydrated with 5% dextrose at 60 °C for 30 min, and then sonicated for 12 min at 4 °C by using a pulse 3/2 s on/off at a power output of 60 W (VCX130, Sonics & Materials Inc). The size, zeta potential and PDI, and morphology were determined by DLS and Cryo-EM, respectively. The molar ratio of Chol

% was calculated according to the following Eq. (1) or Eq. (2):

$$= \frac{\text{mole of(SM} - \text{Chol conjugate)}}{\text{mole of SM} + 2 \times \text{mole of(SM} - \text{Chol conjugate)}} \times 100\% \quad (1)$$

$$= \frac{\text{mole of SMLs}}{0.5 * \text{mole of single chain lipid} + 1.5 \times \text{mole of SMLs}} \times 100\% \quad (2)$$

### Differential scanning calorimetry (DSC)
MicroCal VP-capillary DSC (Malvern Panalytical) was utilized to measure the DSC data. By using DI water as the reference samples, the program of temperature range was set up as 10-90 °C at the rate of 60 °C/h. To measure the DSC data, the lipid film was generated as above and hydrated in DI water (2 mg lipid/mL) under 60 °C for 0.5 h via fitful vortex and then sonicated for 12 min at 4 °C. After that, the liposomes were placed in a water bath at ambient temperature of ~25 °C for 30 min. After degassing, 400 μL of liposome solution was extracted and subjected to DSC analysis. To convert the raw data into molar heat capacity (MHC), VPViewer 2000 and Microcal (LLC Cap DSC Version: Origin70-L3) package software were utilized to collect and analyze the data, respectively.

### Loading calcein into Lipo
The encapsulation of calcein into Lipo followed the reported method[15]. Briefly, calcein (4 mmol, 2.49 g) was dissolved in Tris-HCl buffer (10 mM, 6 mL, pH 7.5,) following adding 50% sodium hydroxide (13.2 mmol, 695 μL). Afterwards, this stock solution was loaded into a Sephadex LH-20 column and eluted using Tris-HCl buffer (10 mM, pH 7.5). The pooled fraction of calcein's concentration was evaluated by measuring the absorbance (494 nm) of the diluted samples at pH 9. Free phospholipids (SM, HSPC, SPC, DSPC, or DOPC) Chol, SMLs (PChcPC, PChemsPC, OChemsPC, and DChemsPC, eq. 40 mol % Chol) and/or SM-Chol conjugate (eq. 40 mol % Chol) were dissolved in ethanol with a 100 mL round bottom glass flask. The solvent was evaporated under a rotatory evaporator (RV 10 digital, IKA) to generate a thin film, which was further dried under an ultra-high vacuum (MaximaDry, Fisherbrand) for 0.5 h. The film was then hydrated with purified calcein (56 mM) solution at 60 °C for 30 min and then sonicated for 12 min at 4 °C by using a pulse 3/2 s on/off at a power output of 60 W (VCX130, Sonics & Materials Inc). The unencapsulated calcein was removed by a PD-10 column (Sephadex G-25, GE Healthcare) using the corresponding isosmotic eluent as eluent.

### Leakage triggered by osmotic stress
The investigation of osmotic stress-induced leakage was performed following the reported method[15]. Briefly, free phospholipids, Chol, SMLs (PChcPC, PChemsPC, OChemsPC, and DChemsPC), and/or SM-Chol conjugate at eq. 40 mol % Chol were dissolved in ethanol with a 100 mL round bottom glass flask. The solvent was evaporated under a rotatory evaporator (RV 10 digital, IKA®) to generate a thin film, which was further dried under ultra-high vacuum (MaximaDry, Fisherbrand) for 0.5 h. The film was hydrated using a mixed solution (56 mM calcein, 10 mM Tris, and 711 mM NaCl), at 60 °C for 30 min, and then sonicated for 12 min at 4 °C by using a pulse 3/2 s on/off at a power output of 60 W (VCX130, Sonics & Materials Inc). The unencapsulated calcein was removed by using a PD-10 column (Sephadex G-25, GE Healthcare) with isosmotic buffer (50 mM HEPES, 775 mM NaCl) as eluent. Lipo (SMLs (PChcPC, PChemsPC, OChemsPC, and DChemsPC), SM/Chol, HSPC/Chol, SPC/Chol, DSPC/Chol or DOPC/Chol with eq. 40 mol % Chol were used as controls. Different osmotic concentration solutions were made by mixing the calcein free isosmotic buffer (50 mM HEPES, 775 mM NaCl, set as 1600 mOsm) and a 50 mOsm dilution buffer (50 mM HEPES). Afterwards, Lipo were placed in solutions with varied osmotic concentrations through mixing Lipo (10 μL) with testing

buffer (990 μL) at 37 °C. After 5 min equilibration, fluorescence signal (excitation: 494 nm; emission: 517 nm) was detected using a SpectraMax M3 reader (SoftMax Pro (v. 7.1.0), Molecular Devices). Via lysing the Lipo using 10% Triton X-100 (100 μL), the total calcein in the Lipo was obtained. The fluorescence of which was determined and set as $F_{100\%}$. The fluorescence intensity of the sample in various osmotic concentrations and in the isosmotic buffer was set as $F_{sample}$ and $F_{blank}$, respectively. The fraction of calcein left in Lipo after osmotic stress-induced leakage was calculated as follows:

$$= 1 - \frac{F_{sample} - F_{blank}}{F_{100\%} - F_{blank}} \qquad (3)$$

### Fetal bovine serum (FBS)-induced leakage
Calcein-laden Lipo was prepared as described above. Lipo (SMLs (PChcPC, PChemsPC, OChemsPC and DChemsPC), SM/Chol, HSPC/Chol, SPC/Chol, DSPC/Chol or DOPC/Chol) containing eq. 40 mol % Chol were used as controls. Lipo samples (an aliquot of 50 μL) were diluted by 30% FBS to reach 2 mL in volume, which were subsequently placed in clean tubes and incubated at 37 °C. At various time points, and the remaining calcein portion in the Lipo was assessed by monitoring the fluorescence intensity as depicted above.

### Preparation of VCR/Lipo-SM-Chol
The remote loading of VCR into Lipo was accomplished according to the reported method[53,81]. Briefly, SM, Chol, PChcPC and/or SM-Chol conjugate with 5 mol % DSPE-PEG$_{2K}$ (for Lipo-SM/Chol, 40 mol % Chol; for Lipo-PChcPC, 64.4 mol %; for Lipo-SM-Chol, 35 mol %) were dissolved in ethanol in a 100 mL round bottom glass flask. The solvent was evaporated under a rotatory evaporator (RV 10 digital, IKA®) to produce a thin film, which was further dried under ultra-high vacuum (MaximaDry, Fisherbrand) for 0.5 h. The lipid film was then hydrated with citrate buffer (300 mM, pH = 4.0) at 60 °C for 30 min, and was then sonicated for 12 min at 4 °C by using a pulse 3/2 s on/off at a power output of 60 W (VCX130, Sonics & Materials Inc). The unloaded citrate buffer was removed by running through a PD-10 column (Sephadex G-25, GE Healthcare) using HBS buffer (20 mM HEPES, 150 mM NaCl, pH 7.5) as the eluent. The remotely encapsulate VCR, citrate buffer-laden Lipo-SM-Chol was incubated with 2 mg/mL VCR (VCR/total lipids = 0.1/1 (w/w)) at 60 °C for 15 min. Afterwards, the samples were left at 4 °C for 30 min; and the unencapsulated VCR was removed by running through a PD-10 column using HBS buffer as eluent. The size, zeta potential PDI, morphology, and drug content of the VCR/Lipo-SM-Chol were determined by DLS, Cryo-EM, and HPLC, respectively. The VCR drug loading capacity [DLC, Eq. (4)] and drug loading efficiency [DLE, equation (5)] were calculated using the formulas shown below:

$$= \frac{\text{weight of encapsulated drug}}{\text{weight of(total lipids + encapsulated drug)}} \times 100\% \qquad (4)$$

$$= \frac{\text{weight of encapsulated drug}}{\text{weight of input drug}} \times 100\% \qquad (5)$$

### Cryo-EM
Liposomal suspensions (eq. 35% molar ratio of Chol in the bilayer for the liposomes; ~2.0 mg VCR/mL for loading VCR, ~9% VCR DLC) were prepared for imaging by applying 3 microliters to the surface of a C-Flat 1.2/1.3 engineered TEM grid (Protochips, Morrisville NC.) immediately followed by either a 3 or 6 s blot at 100% RH in a FEI Vitrobot (Hillsboro, OR.) prior to rapid emersion into liquid nitrogen cooled liquid ethane. Grids were transferred into a Phillips TF20 (Eindhoven NL.) operating at 120 KeV with a Gatan CT3500 side entry

cryoholder (Pleasantville, CA.) maintained at -180 °C. Images were recorded on a TVIPS XF416 CMOS camera at the indicated and measurements were performed within the EMMenu software package provided by TVIPS (Gauting, DE) for the operation of the XF416 camera.

### Preparation of DOX/Lipo-SM-Chol
Phospholipid, Chol and DSPE-PEG$_{2K}$ at the indicated molar ratio in Fig. 5a were completely dissolved by ethanol in a 100 mL round bottom glass flask. The thin lipid film was generated through evaporating the organic solvent via a rotatory evaporator (RV 10 digital, IKA®) under ultra-high vacuum (MaximaDry, Fisherbrand) for half an hour. $(NH_4)_2SO_4$ buffer (125 mM) was added into the flask to hydrate the lipid film under 60 °C and rotated for half an hour. The suspension solution was transferred to a tube and sonicated via a probe under 4 °C through a pulse 3/2 s on/off at a power output of 60 W (VCX130, Sonics & Materials Inc). To remove the unencapsulated $(NH_4)_2SO_4$, the primary liposomal solution was gone through a PD-10 column (Sephadex G-25, GE Healthcare) via PBS buffer as eluent. Afterwards, the $(NH_4)_2SO_4$ laden liposomes were incubated with 6 mg/mL DOX by the ratio of DOX/total lipid = 0.1/1 (w/w) under 60 °C for 1 h. The liposomal solutions were transferred to an ice bath and cooled to ~4 °C for half an hour. To get rid of the unloaded DOX, the liposomes were passed through a PD-10 column (Sephadex G-25, GE Healthcare) via PBS buffer as eluent. DLS and HPLC were utilized to measure the size, zeta potential, PDI, and drug content of the liposome samples, respectively. The DLC [Eq. (4)] and DLE [equation (5)] of DOX in the liposomes were calculated using the formulas as above:

### Preparation of IRI/Lipo-SM-Chol
The remote loading of IRI into Lipo was prepared based on the reported method[63]. Sucrose octasulfate (TEA$_8$SOS) were obtained from commercially available Na$_8$SOS based on ion-exchange chromatography using the resin (Dowex 50Wx8-200) in the H$^+$ form, which was immediately titrated by neat triethylamine (TEA) to reach a pH of 5.5-6.0. The TEA$_8$SOS concentration was then adjusted to ~200 mM. SM, Chol, SMLs and/or SM-Chol conjugate with 35 mol % Chol and 5 mol % DSPE-PEG$_{2K}$ (for Lipo-SMLs, eq. 64.4 mol% Chol, except eq. 97.4 mol% Chol for IRI/Lipo-DChemsPC since two Chol molecules were consisted in DChemsPC) were dissolved in ethanol in a 100 mL round bottom glass flask. The solvent was evaporated under a rotatory evaporator (RV 10 digital, IKA®) to generate a thin film, which was further dried under ultra-high vacuum (MaximaDry, Fisherbrand) for 0.5 h. The film was then hydrated with TEA$_8$SOS buffer at 60 °C for 30 min, followed by sonication for 12 min at 4 °C by using a pulse 3/2 s on/off at a power output of 60 W (VCX130, Sonics & Materials Inc). The unloaded TEA$_8$SOS was removed by a PD-10 column (Sephadex G-25, GE Healthcare) with HEPES-buffered dextrose (5% dextrose, 5 mmol/L HEPES, pH 6.5) as the eluent. For IRI remote loading, TEA$_8$SOS-loaded Lipo-SM-Chol was incubated with 10 mg/mL IRI (IRI/total lipid = 0.1/1 (w/w)) at 60 °C for 30 min. After cooling the samples down at 4 °C for 30 min, the unencapsulated IRI was removed by running through a PD-10 column using HEPES-buffered saline (145 mM NaCl, 5 mM HEPES, pH 6.5) as the eluent. The size, zeta potential and PDI, morphoilogy, and drug content of the IRI/Lipo-SM-Chol were determined by DLS, cryo-EM, and HPLC, respectively. The IRI DLC [Eq. (4)] and DLE [equation (5)] were calculated based on the formulas described above.

### Preparation of DEX/Lipo-SM-Chol
SM, Chol, PChcPC and/or SM-Chol conjugate with 35 mol % Chol and 5 mol % DSPE-PEG$_{2K}$ (for Lipo-PChcPC, eq. 64.4 mol % Chol) were dissolved in ethanol with a 100 mL round bottom glass flask. DEX (5% (w/w) of the total lipids) was well mixed in the solution. The solvent was evaporated under a rotatory evaporator (RV 10 digital, IKA®) to generate a thin film, which was further dried under ultra-high vacuum

(MaximaDry, Fisherbrand) for 0.5 h. The film was hydrated with 5% dextrose at 60 °C for 30 min, and then sonicated for 12 min at 4 °C by using a pulse 3/2 s on/off at a power output of 60 W (VCX130, Sonics & Materials Inc). The unencapsulated DEX was removed by a PD-10 column (Sephadex G-25, GE Healthcare) by using 5% dextrose as the eluent. The size, zeta potential PDI, morphology, and drug content of the DEX/Lipo-SM-Chol were determined by DLS, cryo-EM, and HPLC, respectively. The DEX DLC [Eq. (4)] DLE [equation (5)] was calculated using the formula provided above.

## Preparation of siRNA/LNP

To efficiently encapsulate siRNA, ionizable lipid DLin-MC3-DMA (DMA, WuXi App Tec) used in FDA-approved siRNA nanotherapeutic, Onpattro[1], was also used. Briefly, DMA, DSPC, Chol, and PEG$_{2K}$-C-DMG (BOC Sciences, NY, USA) at the molar ratio of 49.3/10.2/39.0/1.5 as in Onpattro were used as the control lipid bilayer and dissolved in ethanol, this same lipid ratio was used for LNP-DMA/SM/Chol. For LNP-DMA/SM-CSS-Chol, the lipid molar ratio is 49.3/10.2/39.0/1.5 (DMA/SM/SM-CSS-Chol/PEG$_{2K}$-C-DMG), for LNP-DMA/PChcPC, the lipid molar ratio is 49.3/10.2/39.0/1.5 (DMA/1-palmitoyl-2-hydroxy-sn-glycero-3-phosphocholine /PChcPC/PEG$_{2K}$-C-DMG). The lipid solution was added subsequently in 1.85-fold volumes of citrate buffer (25 mmol/L, pH 4.0) containing P-gp siRNA with vigorous stirring. The siRNA sequence (5'-3': GGAUCCAGUCUAAUAAGAAtt; Antisense: UUCUUAUUAGACUGGAUCCtg) targeting the *Abcb1a* gene (Assay ID 156774, #AM16704, Ambion, USA) was included at a ratio of 0.056 mg/μmol (N/P = 6:1) to total lipids. The solution was incubated at room temperature for 30 min and subsequently dialyzed overnight in 5% dextrose at 4 °C. For serum stability analysis[11], naked siRNA or different siRNA-loaded LNP were mixed with FBS (v:v = 1:1) and incubated at 37 °C for various times (0, 1, 2, 4, 8, 12, and 24 h). The siRNA stability was then analyzed by 1% agarose gel electrophoresis (100 V, 30 min).

## Therapeutic efficacy of drug-laden Lipo

**Subcutaneous SUDHL4 DLBCL xenograft tumor model.** CB17/Icr-*Prkdc^scid^*/IcrIcoCrl mice (n = 5 mice, female) were subcutaneously injected with $1 \times 10^7$ SUDHL4 cells in 100 μL RPMI-1640 medium with Matrigel (Corning, Discovery labware Inc.) (3/1, v/v). When tumors grew to ~200 mm³ in size, the mice received one i.v. dose of 5% dextrose (vehicle control), or VCR/Lipo (2 mg VCR/kg). Tumor development, mice body weight and survival were closely monitored as indicated. In an independent study, after the tumor-bearing mice received the same treatments, the tumors were dissected and subjected to immunofluorescence (IF), immunohistochemistry (IHC) and TUNEL staining.

## Orthotopic metastatic 4T1-Luc2 (luciferase-expressing) TNBC tumor model

To establish the 4T1-Luc2 orthotopic model[82], BALB/c mice (n = 5 mice, female) were anesthetized by isoflurane. The hair/fur in the abdominal area of mice were removed by a shaver. Then the surgical area underwent three alternating scrubs of betadine/povidone iodine followed by 70% ethanol. Buprenorphine SR (1.0 mg/kg) was subcutaneously administered to mice before surgery. Then, a ~1 cm abdominal incision was created with a sterile disposable scalpel and the 4th mammary fat pad was exposed. A total of $2 \times 10^5$ 4T1-Luc2 cells in 50 μL of RPMI-1640 medium with Matrigel (Corning, Discovery labware Inc.) (3/1, v/v) were inoculated into the 4th mammary fat pad by using a 26-gauge needle (BD precisionGlide™). After sterilizing the injection site with 70% ethanol (to kill cancer cells that may have leaked out), the mammary fat pad was then replaced into the s.c. cavity. The skin was then closed with wound clip (BD Diagnostic). Surgical glue was also applied to allow good apposition of skin. During and after surgery, animals were placed on the heating pad and were closely monitored until ambulatory; and then mice were returned to a clean cage. At indicated time points, tumor size was measured by a digital caliper, and tumor burden of a whole mouse body was determined by bioluminescence radiance intensity using Lago optical imaging after mice were intraperitoneally injected with 150 mg/kg *D*-Luciferin (GoldBio, MO, USA). When tumors grew to ~200 mm³ in size on day 15, the mice received one dose of intravenously administered 5% dextrose (vehicle control), DOX/Lipo (15 mg DOX/kg). On day 35, following injection of *D*-Luciferin, mice were dissected, and lungs were quickly obtained and then subject to ex vivo Lago imaging to investigate the tumor metastasis. The tumors were dissected and subjected to IHC and TUNEL staining.

## Lung inflammation

To establish the lipopolysaccharide (LPS)-induced lung inflammation model[10,83], Balb/c mice (n = 5 mice, female) were first anesthetized with isoflurane and kept at a heating pad (37 °C). The neck of mice was extended in a 90° angle relative to the pad, and the tongue of the mice was held with forceps to straighten the throat to facilitate intubation conditions. We then cut the No. 22 gauge (G) catheter to a length of 20 mm and gently inserted the catheter vertically along the base of the tongue and about 10 mm into the trachea. LPS (30 μL) was then slowly injected into the trachea using a syringe and the tube was slowly removed after injection. The upper body of the mouse was kept upright for 30 s to avoid fluid leakage from the trachea. In sham animals, 30 μL sterile 5% dextrose was injected intratracheally instead of LPS. The mice were kept on a heating pad (37 °C) until full consciousness was restored. 6 h after the challenge, 1 mg/kg (200 μL, 100 μg/mL) of free DEX, or DEX/Lipo was injected intravenously to mice. 12 h after treatment, lungs were collected and cut into two parts, and weighted. One part of the lung tissues were homogenized and the homogenates were centrifuged at 6000 g, and then the supernatants were collected for measuring the IL-6 levels using a mouse IL-6 ELISA kit (Abcam222503), TNF-α levels using mouse TNF-α ELISA kit (Thermo Fisher, BMS607-3) and IL-1β levels using mouse IL-1β ELISA kit (Thermo Fisher, BMS6002) according to the manufacturer's protocol. Another part of lung tissues was fixed in 4% paraformaldehyde overnight and then sent to the Tissue Acquisition and Cellular/Molecular Analysis Shared Resource (TACMASR) at UArizona Cancer Center (UACC) for H & E staining analysis. Histology images were obtained using a Leica DMI6000B microscope, with a Leica DFC450 color camera and the Leica LAS X 3.7 software. The images were analyzed by LAS X 3.7 software (v. 3.7.3.23245).

## Subcutaneous CT26 CRC tumor model

BALB/c mice (n = 5 mice, female) were subcutaneously injected with $1 \times 10^6$ CT26 cells in 100 μL of serum-free RPMI-1640 medium. When tumors grew to ~100 mm³ in size on day 10, the mice were intravenously administered by 5% dextrose (vehicle control), IRI/Lipo-PChcPC (40 mg IRI/kg), IRI/Lipo-CSS-Chol (40 mg IRI/kg), combination of IRI/Lipo-PChcPC (40 mg IRI/kg) and siRNA/LNP-DMA/PChcPC (200 μg siRNA/kg) or combination of IRI/Lipo-CSS-Chol (40 mg IRI/kg) and siRNA/LNP-DMA/SM-CSS-Chol (200 μg siRNA/kg) every 2 days for 3 times. Tumor growth and mice body weight were closely monitored as indicated. 24 h after the last drug injection, the mice were euthanized and tumor tissues were collected, weighed, and then homogenized in acidified methanol (0.075 M HCl, 900 μL per 100 mg tissue) prior to the IRI content measurement by the established HPLC method (Supplementary Fig. 39). For P-gp gene (full gene name: ATP-binding cassette, sub-family B (MDR/TAP), member 1 A (*Abcb1a*)) knockdown, BALB/c mice (n = 3 mice) were subcutaneously injected with $1 \times 10^6$ cells per mouse in 100 μL of serum-free RPMI-1640 medium. When tumors reached ~100 mm³ in size, siRNA/LNP-DMA/DSPC/Chol, siRNA/LNP-DMA/PChcPC, siRNA/LNP-DMA/SM/Chol, siRNA/LNP-DMA/SM-CSS-Chol was intravenously injected to mice at 200 μg siRNA/kg every 2 days for 3 times. On day 15, the mice were euthanized, and tumor

tissues were isolated and processed for qRT-PCR (the reverse transcription polymerase chain reaction (RT-PCR) and quantitative real-time PCR (qPCR) combined technique) to measure the P-gp mRNA levels (*Abcb1a* mouse qPCR primer pair (Gene ID, 18671), forward sequence: TCCTCACCAAGCGACTCCGATA, reverse sequence: ACTTGAGCAGCATCGTTGGCGA, #MP201132, OriGene, USA) following the published method[36].

## Statistical analysis

The level of significance in all statistical analyzes was set at $P < 0.05$. Data are presented as mean ± s.d. and were analyzed using the two-tailed, unpaired Student's *t*-test for two groups or one-way analysis of variance (ANOVA) for three or more groups followed by Tukey's multiple comparisons test using Prism 8.0 (GraphPad Software). Kaplan–Meier survival curves were compared with the log-rank Mantel–Cox test.

## Reporting summary

Further information on research design is available in the Nature Portfolio Reporting Summary linked to this article.

## Data availability

All the data supporting the findings of this study are available within the article and its Supplementary Information. The full image dataset is available from the corresponding author upon request. Source data are provided with this paper.

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

## Acknowledgements

This work was supported in part by a Startup Fund from the College of Pharmacy at The University of Arizona (UArizona) and a PhRMA Foundation Faculty Starter Grant in Drug Delivery, and by National Institutes of Health (NIH) grants (R35GM147002 and R01CA272487) to J.L. We acknowledge the use of Mass Spectrometry in Analytical and Biological Mass Spectrometry Core Facility at the UArizona BIO5 Institute; the UArizona Translational Bioimaging Resource Core for the Lago imaging; UArizona University Animal Care Pathology Services for the serum chemistry and hematological counts; The W.M. Keck Center for Surface and Interface Imaging at UArizona for AFM; Tissue Acquisition and Cellular/Molecular Analysis Shared Resource (TACMASR) at UArizona Cancer Center (UACC) for the H&E staining and confocal laser scanning microscopy; we thank Patty Jansma, manager of the Office of Research, Innovation and Impact's Imaging Core-Optical Core Facility at the University of AZ for providing training and support; and Automated Biological Calorimetry facility at The Pennsylvania State University's Huck Institutes of the Life Sciences for MicroCal VP-capillary DSC; and Arizona State University's John Cowley Center for Hight Resolution Electron Microscopy (the specific instrumentation used was supported by the NSF, MRI grant NSF1531991) for Cryo-EM. We are also grateful for Gregory Beatty (University of Pennsylvania) for providing the KPC-Luc cells; Catherine Smith (UArizona) for providing the SU-DHL-4 cells; and Wei-guo Han (UArizona) for assistance in PCR assay.

## Author contributions

J.L. conceived and supervised the project. J.L., Z.W. and W.L. designed the experiments, analyzed the data, and wrote the manuscript. Z.W. synthesized the SM-Chol conjugates, prepared the VCR/Lipo, IRI/Lipo, DEX/Lipo, MP-P/Lipo, and DOX/Lipo, and performed pharmacokinetics and efficacy studies in MC38, SU-DHL-4, KPC-Luc, and 4T1-Luc2 tumor models. W.L. prepared the siRNA/LNP and carried out the efficacy studies in lung inflammation and CT26 tumor models, as well as assisted the studies in KPC-Luc and 4T1-Luc2 tumor models. Z.W. and W.L. performed the physicochemical characterizations, leakage, and Chol exchange studies of nanotherapeutics and HPLC drug analysis. Y.J., J.P., and K.M.G. assisted in the SM-Chol synthesis, liposome preparation, and sample preparation for HPLC analysis. Z.W., X.W. and Q-Y.Z. performed the stability assay by LC-MS/MS. All the authors discussed the results and commented on the manuscript.

## Competing interests

J.L., Z.W. and W.L. have applied for patents related to this study. The remaining authors declare no competing interests.
