## [Peer Review File · Nature Communications]

Reviewers' Comments:

Reviewer #1:

Remarks to the Author:

The revised manuscript by Wang et al. has largely addressed the previous scientific concerns raised in the previous review. In this context, the authors should be commended for synthesizing a brand new lipid to test in their experiments, as requested by one reviewer. However, the previous comment about the figures containing too much data is still a concern and I find it difficult/impossible to digest the overwhelming amount of data in this manuscript; many scientists would consider this a good decade's worth of work. In that sense, I believe that the results would be significantly more digestible if they were separated into multiple manuscripts, perhaps based on the different therapeutic molecules or the different animal models. In this sense, I concur with the prior comments regarding the main figures containing too much data, and the manuscript also includes "extended data" and over 30 supplementary figures! This is an impressive amount of work, but it is simply too much for a reader to comprehend. I also want to echo previous comments regarding the poor presentation (due mostly to awkward wording) and the fact that this highly specialized manuscript is a better fit for a drug delivery or lipid-focused journal. A couple of minor comments are included below.

1. "Lipo" is an odd abbreviation for liposome, and I find it confusing.
2. While cholesterol is rapidly transferred, many liposomal drugs contain cholesterol, so it is not clear that this transfer is a significant problem that must be overcome.

Reviewer #2:

Remarks to the Author:

I recommend to accept this Transfer article for publication in Nature Communications. The major points were addressed from the previous review. It has been improved and likely better matches the scope and level of Nature Communications with abundant data in cancer therapy for new delivery vehicles. I suggest to accept in the present form.

Reviewer #1: (Remarks to the Author):

The revised manuscript by Wang et al. has largely addressed the previous scientific concerns raised in the previous review. In this context, the authors should be commended for synthesizing a brand new lipid to test in their experiments, as requested by one reviewer. However, the previous comment about the figures containing too much data is still a concern and I find it difficult/impossible to digest the overwhelming amount of data in this manuscript; many scientists would consider this a good decade's worth of work. In that sense, I believe that the results would be significantly more digestible if they were separated into multiple manuscripts, perhaps based on the different therapeutic molecules or the different animal models. In this sense, I concur with the prior comments regarding the main figures containing too much data, and the manuscript also includes "extended data" and over 30 supplementary figures! This is an impressive amount of work, but it is simply too much for a reader to comprehend. I also want to echo previous comments regarding the poor presentation (due mostly to awkward wording) and the fact that this highly specialized manuscript is a better fit for a drug delivery or lipid-focused journal.

Response: Thank you for the comment. To improve the readability and allow the ease of the digestion for readers, we have moved the Extended Data Fig. 1, 2 to Supplementary Figure 37 and 42 in the supplemental information, respectively.

A couple of minor comments are included below.

Q1. "Lipo" is an odd abbreviation for liposome, and I find it confusing.

Response: Thank you for the comment. We have presented the abbreviation for liposome as "Lipo" in the very first place of the introduction. We believe that reasonable readers will be able to fathom the meaning of "Lipo" after reading the introduction.

Q2. While cholesterol is rapidly transferred, many liposomal drugs contain cholesterol, so it is not clear that this transfer is a significant problem that must be overcome.

Response: Thank you for your comment. The significance of addressing the cholesterol transfer in liposomal drug delivery has been highlighted in the introduction as follows:

"While most FDA-approved liposomal nanotherapeutics can improve pharmacokinetics and ameliorate side effects, improvements in therapeutic efficacy and overall survival are limited even for the best nanoformulations and completely missing for majority^{5,6}, underscoring the urgent need of an improved platform for enhanced therapeutic delivery. Lipid bilayers with a high percentage of cholesterol (Chol) are generally more stable than those with less Chol¹¹. Nevertheless, Chol can be readily transferred between biomembranes and lipoproteins under physiological conditions¹²⁻¹⁴,

which sabotages liposomal stability and results in premature contents leakage, subsequent fast blood clearance and unwanted systemic adverse effects, resulting in disappointing therapeutic efficacy in clinic^{5,6}.”

To tackle this key bottleneck in liposomal drug delivery, an improved lipid bilayer that forms Lipo but cannot shuttle between biomembranes to cement drug packaging and therapeutic delivery is ideal. Herein, via systemic structure activity relationship screening we demonstrated that SM-Chol with a disulfide bond and longer linker outperformed other counterparts and conventional phospholipids/Chol mixture systems on blocking Chol transfer and payload leakage, increased maximum tolerated dose of vincristine while reducing systemic toxicities, improved pharmacokinetics and tumor delivery efficiency, and enhanced antitumor efficacy in SU-DHL-4 diffuse large B-cell lymphoma xenograft model. Furthermore, SM-Chol improved therapeutic delivery of structurally diversified therapeutic agents (irinotecan, doxorubicin, dexamethasone) or siRNA targeting multi-drug resistant gene (p-glycoprotein) in late-stage metastatic orthotopic KPC-Luc pancreas cancer, 4T1-Luc2 triple negative breast cancer, lung inflammation, and CT26 colorectal cancer animal models compared to respective FDA-approved nanotherapeutics or lipid compositions. Thus, SM-Chol represents a promising platform for universal and improved drug delivery.

Reviewer #2: (Remarks to the Author):

I recommend to accept this Transfer article for publication in Nature Communications. The major points were addressed from the previous review. It has been improved and likely better matches the scope and level of Nature Communications with abundant data in cancer therapy for new delivery vehicles. I suggest to accept in the present form.

Response: Thank you!

We hope that all the raised concerns have been addressed in a satisfactory manner and that this revised manuscript is now acceptable for publication in Nature Communications. Please let us know should more information is needed. Thank you very much for your consideration.

We wish you and your loved ones are safe and healthy during this troubling time.

Sincerely,

Jianqin Lu, BPharm, PhD

Assistant Professor & Director

Pharmaceutics & Pharmacokinetics Track

Department of Pharmacology & Toxicology

R. Ken Coit College of Pharmacy

NCI-designated UArizona Comprehensive Cancer Center

BIO5 Institute & Southwest Environmental Health Sciences Center

The University of Arizona, Tucson, Arizona

Office: 520-626-1786; Email: lu6@arizona.edu